# Identification of Potential miRNA–mRNA Regulatory Network Associated with Growth and Development of Hair Follicles in Forest Musk Deer

**DOI:** 10.3390/ani13243869

**Published:** 2023-12-15

**Authors:** Wen-Hua Qi, Ting Liu, Cheng-Li Zheng, Qi Zhao, Nong Zhou, Gui-Jun Zhao

**Affiliations:** 1College of Biological and Food Engineering, Chongqing Three Gorges University, Chongqing 404100, China; wenhuaqi357@163.com (W.-H.Q.); liuting@stumail.sanxiau.edu.cn (T.L.); zhaoqi@stumail.sanxiau.edu.cn (Q.Z.); 2Sichuan Institute of Musk Deer Breeding, Chengdu 611830, China; zcl19861106@163.com; 3Chongqing Institute of Medicinal Plant Cultivation, Chongqing 408435, China

**Keywords:** hair follicle, transcriptome, miRNA–mRNA network, signal pathway analysis, RT-qPCR, forest musk deer

## Abstract

**Simple Summary:**

The forest musk deer (FMD), an endangered artiodactyl species, produces valuable musk secretions with significant medicinal and economic importance. Hair follicles (HFs) play a crucial role in skin rebirth. The periodic development and growth of HFs are regulated by melatonin, genes, and miRNAs. RNA-seq technology was used to find unigenes and miRNAs related to HF growth and development in the HFs of FMD at the anagen stage and catagen stage. The results showed that some key genes and miRNAs affected HF growth and development, which were significantly enriched in the related signaling pathway of HF development and growth. This study provides potential miRNA–mRNA regulation mechanisms for HF growth and development.

**Abstract:**

In this study, sRNA libraries and mRNA libraries of HFs of FMD were constructed and sequenced using an Illumina HiSeq 2500, and the expression profiles of miRNAs and genes in the HFs of FMD were obtained at the anagen and catagen stages. In total, 565 differentially expressed unigenes (DEGs) were identified, 90 of which were upregulated and 475 of which were downregulated. In the BP category of GO enrichment, the DEGs were enriched in the processes related to HF development and differentiation, including the hair cycle regulation and processes, HF development, skin epidermis development, regulation of HF development, skin development, the Wnt signaling pathway, and the BMP signaling pathway. Through KEGG analysis it was found that DEGs were significantly enriched in pathways associated with HF development and growth. A total of 186 differentially expressed miRNAs (DEmiRNAs) were screened (*p* < 0.05) in the HFs of FMD at the anagen stage vs. the catagen stage, 33 of which were upregulated and 153 of which were downregulated. Through DEmiRNA–mRNA association analysis, we found DEmiRNAs and target genes that mainly play regulatory roles in HF development and growth. The enrichment analysis of DEmiRNA target genes revealed similarities with the enrichment results of DEGs associated with HF development. Notably, both sets of genes were enriched in key pathways such as the Notch signaling pathway, melanogenesis, the cAMP signaling pathway, and cGMP-PKG. To validate our findings, we selected 11 DEGs and 11 DEmiRNAs for experimental verification using RT-qPCR. The results of the experimental validation were consistent with the RNA-Seq results.

## 1. Introduction

Musk deer (*Moschus* spp.) have five recognized species, one of which is the forest musk deer (Moschus berezovskii). Musk deer are endemic to Asia and were distributed in at least 13 countries in Asia [1]. The decline in musk deer populations to near extinction has attracted much attention from governments and related international organizations around the world. All species of musk deer are listed in the World Conservation Union IUCN Red List of Threatened Species [2]. Male musk deer possess a specialized gland for producing musk secretions that are highly valuable in animal products. Musk has been used for perfumes and medicines in Asian countries for many years. Musk deer have been illegally poached for their musk. This has led to a dramatic decrease in the musk deer population. Artificial breeding has played a significant role in the conservation of wild populations of musk deer and in the sustainable use of musk resources. So far, the forest musk deer (FMD) is the largest breeding population of musk in China.

Hair follicles (HFs) are unique appendage organs found in mammals, serving as vital components of skin appendages and playing a pivotal role in skin rebirth [3]. The growth and development of HFs are regulated by various factors such as melatonin, genes, miRNAs, lncRNA, and signaling pathways [4]. Seasonal variations in melatonin levels could impact the genetic factors that affect HF development. It has been shown that multiple genes, including *BMP4* [5], *Dlx3*, *lef1* [6], *EGF*, *FGF5*, *IGF-1*, *Sox9* [7], and *Wnt10a* [8], promote or inhibit HF growth and development [9,10,11,12,13]. Furthermore, the BMP [14], Eda [15], Shh [16], and TGF-β [17] signaling pathways actively participate in HF growth and development. It has been indicated that the expression level of melatonin-associated Hoxc13 significantly increases, which positively regulates the relevant gene expression in the HF development of cashmere goat [18].

MicroRNAs (miRNAs) are small endogenous RNAs that play pivotal regulatory roles in biological processes via targeted genes [19]. Previous studies have demonstrated that miRNAs could competitively bind to 3’ untranslated regions (3’UTR) of mRNAs to suppress the expression of target genes [20]. Numerous miRNAs have been implicated in the regulation of crucial cellular activities, such as cell development, proliferation, and apoptosis, particularly the occurrence and progression of disease [21]. Recent studies have shown that HF development and growth may also be affected by miRNAs. miRNAs and their target genes constitute complex regulatory networks, which play a crucial role in the development and growth of HFs. miRNAs are expressed in various types of HF cells, many of which have been confirmed to have specific expression patterns within HF cells [22]. Although miR-125b [23] and miR-205 [24] are all expressed in HF stem cells, their functions differ significantly. miR-125b can suppress HF stem cell differentiation by competitive binding of *Vdr* and *Blimp1* [23], while miR-205 positively regulates HF stem cell proliferation by competing to bind 3’UTR of *INPP1b*, *INPP4b*, *PHLDA2*, and *FRK* [24]. The expression of miR-24 is abundant in the inner root sheath, which promotes HF keratinocyte differentiation by competitive binding of *TCF-3* [25]. Overexpression of miR-137 can lead to hair depigmentation by targeting *Mitf*, *Tyr*, *Trp1*, and *Trp2*, which are involved in melanocyte development and melanogenesis [26]. miR-31 is expressed in the HF matrix and the outer root sheath, which inhibits anagen progression and hair shaft differentiation by binding to *KRT16*, *KRT17*, *FGF10*, and *DLX3* [27]. miR-214 inhibits hair growth, which is mediated by activating the Wnt and Shh signaling pathways [28].

Profile analysis of gene expression helps discover and predict gene functions. Differentially expressed unigenes (DEGs) of HF development have been obtained, and DEGs associated with HF growth and development have been found, all of which could be utilized in RNA-Seq analysis. miRNAs perform important functions in HF development and growth, and a lack of miRNAs may lead to abnormal development [29]. Therefore, it is crucial to study HF development and growth via miRNA–mRNA analysis. However, there is no available transcriptome information about the growth and development of FMD. The molecular regulation of HF development and growth has not yet been researched in moschidae, and further investigations are necessary to detect the specific molecular mechanisms. Therefore, we sequenced the transcriptome and small RNA across two important HF development stages (i.e., anagen and catagen stages) from the HFs of 13 FMD. We constructed a regulatory network of DEGs and DEmiRNA target genes in order to reveal their functions in HFs. Additionally, hub DEGs and core DEmiRNAs were verified using RT-qPCR in the HF tissues of FMD. The HFs of FMD showed different color changes at different stages. The 3 cm HFs of FMD were black at the anagen stage, when they contain more melanin. The HFs of FMD became white due to melanin cells undergoing apoptosis at the catagen stage [30]. Investigating the molecular mechanisms of HF development and growth at the various stages of FMD development is therefore crucial. This study’s results will help identify the novel functional genes and miRNAs related to HF or skin development and clarify the molecular mechanisms of HF development in FMD, which will help control hair quality and thus play an important role in maintaining body temperature.

## 2. Materials and Methods

### 2.1. Sample Collection and RNA Extraction, Library Preparation, and Sequencing

In this study, we collected HF samples from 13 FMD at the Chongqing Institute of Medicinal Plant Cultivation in Chongqing, China. The collected HF tissues were divided into two groups based on their developmental stages: the anagen stage group, which consisted of HFs that were black, approximately 3 cm long, and contained higher levels of melanin, and the catagen stage group, which consisted of HFs that stopped producing melanin and exhibited apoptosis in some melanin cells. The HFs of the anagen stage were collected during the hair replacement period in September–October 2021. The HFs at the catagen stage were collected in January–February 2022. At least three biological duplicate samples were collected for each developmental stage. All sample collections and handling procedures were approved by both the Ethics Committee of Chongqing Three Gorges University and the Ethics Committee of the Chongqing Institute of Medicinal Plant Cultivation. After manual anesthesia, the HF tissues were extracted and stored in liquid nitrogen until RNA extraction. Total RNAs were subsequently isolated and purified using the TRIzol reagent. Double-stranded cDNAs were synthesized based on mRNA fragments, and cDNA fragment purification was achieved using AMPure XP beads prior to ligation to poly(A) and sequencing adapters. These HF tissues were sequenced on the Illumina HiSeq 2500 platform. The remaining HF samples were saved at −80 °C.

### 2.2. Analysis of Sequencing Data

Total RNAs of HF tissues of FMD individuals were used to build an mRNA library or an sRNA library for the two developmental stages. According to the method we used in our lab, build an mRNA library or an sRNA library were produced and sequenced using an Illumina HiSeq2500 [31,32]. The analytical methods of DEGs were used in accordance with previous reports from our lab [31,32]. The miRNA-seq reads were filtered, and clean reads were then mapped to the mature cow miRNA database (*Bos taurus* V3.1) [33] in miRBase (v21) as well as to the Rfam database. Novel miRNAs were also predicted using miRDeep2 [34]. Unigenes with Padj < 0.05 and |log2(fold change)| ≥ 2 were considered to be the DEGs. The DEmiRNAs were recognized by threshold [*p* < 0.05 and |log2 (fold change) |>1].

### 2.3. GO Enrichment and KEGG Pathway Analyses

All the DEGs and DEmiRNA target genes were analyzed using the GO and KEGG databases. The GO enrichment analysis was utilized to assess the functions of the DEGs and DEmiRNA target genes. The GO terms with *p* < 0.05 were obviously enriched. In addition, KEGG enrichment analysis was used to explore the pathways in which the DEGs and DEmiRNA target genes participated. The KEGG terms with *p* < 0.05 were also obviously enriched.

### 2.4. DEGs Network or DEmiRNA–mRNA Network Integration

Based on the STRING protein interaction database, interaction confidence ≥0.5 was considered significant. Then, the DEG information was imported into Cytoscape software 3.8.0 for further analysis and visualization. By utilizing Cytoscape, we established a network of interactions among the DEGs in the HF tissues at the anagen stage vs. catagen stage. The hub genes were recognized using Cytoscape, which was also used to show the protein–protein interaction network for DEGs. PicTar, miRanda, and RNAhybrid were used to forecast the target genes of all miRNAs. Then, more than two miRNA target genes were taken as target genes. Subsequently, the negatively co-expressed DEmiRNA and their target gene pairs were screened to construct DEmiRNA–mRNA networks in the HF tissues (PCC < 0.7, *p* < 0.05). Lastly, the interactive networks of DEmiRNAs and target genes were visualized in Cytoscape 3.8.1.

### 2.5. RT-qPCR Validation of DEGs and DEmiRNAs

In order to check the correctness of sequencing data, hub DEGs were verified by RT-qPCR. Reverse transcription of total RNA was carried out using a Universal RT-PCR Kit. The RT-qPCR assays were conducted with a SYBR Green qPCR Mix (Sinogene, Beijing, China) on a PCR analysis system under the following conditions: initial denaturation at 95 °C for 10 min, 40 cycles of denaturation at 95 °C for 20 s, annealing at 60 °C for 30 s, followed by dissociation curve analysis. miRNAs were reverse-transcribed using an MiRNA qPCR Kit (SYBR Green Method). Fluorescent quantitative primers were designed with primer5 (Appendix A). Using GAPDH and U6 snRNA as the reference genes, the expression levels of genes were calculated using the 2^−ΔΔCt^ method. We present the data as the mean ± SE of three experiments. T-tests were used to assess statistical significance (* *p* ≤ 0.05 and ** *p* ≤ 0.01).

## 3. Results

### 3.1. Overview of mRNA Library of HFs

In total, 1,270,042,376 and 635,604,846 raw reads were produced in the HFs at the anagen stage and catagen stage, respectively. After the quality control (QC) of these raw reads, 1,236,923,984 (93.89%) and 623,760,090 (93.78%) clean reads were obtained at the anagen stage and catagen stage, respectively, for further analysis (Appendix A). An average of 11,770 (73.42%) and 12,157 (74.43%) unigenes with FPKM > 0.5 were obtained in the HFs at the anagen stage and catagen stage, respectively (Appendix A). There were 71,167 transcripts, ranging from 132 bp to 101,907 bp, with an N50 value of 4942 bp and a mean length of 2559.9 bp. A detailed overview is given in Table 1 and Figure 1.

### 3.2. DEGs and Functional Enrichment Analysis

In comparison to the HFs at the catagen stage, a total of 565 DEGs among 24,352 expressed unigenes were identified in the HFs at the anagen stage. Among them, 12,654 unigenes were co-expressed in the two stages, and 355 and 593 unigenes were peculiarly expressed only at the anagen and catagen stages, respectively (Figure 2B). A heatmap clustering of the 13 samples of DEGs was performed, which showed that gene and group clusters were significant (Figure 2A). Among them, 90 up- and 475 downregulated DEGs were detected, and 85 and 460 annotated up- and downregulated DEGs were identified (Figure 2B, Appendix A). In order to explore the biological functions of DEGs connected to HF development in FMD during the HF cycle, GO and KEGG analyses were performed at the anagen and catagen stages. In GO enrichment analysis, 913 BP terms were found from the anagen vs. catagen groups (*p* < 0.05; Appendix A). In the BP category, the DEGs were obviously enriched in blood vessel development, vasculature development, positive regulation of multicellular organismal process, regulation of signaling, anatomical structure formation involved in morphogenesis, blood vessel morphogenesis (Appendix A). Multiple terms of these BPs have been reported to be connected to HF development and growth.

A set of significant pathways was found in the KEGG analysis of 565 DEGs of HFs at the anagen stage and catagen stage (Figure 3C; Appendix A). These significant pathways were identified as follows: the PI3K-Akt signaling pathway, Wnt signaling pathway, development and regeneration, melanogenesis, osteoclast differentiation, relaxin signaling pathway, Rap1 signaling pathway, cAMP signaling pathway, growth hormone synthesis, secretion and action, cGMP-PKG signaling pathway, MAPK signaling pathway, and Toll and Imd signaling pathways.

### 3.3. Interaction Network Analysis of DEGs

In our study, a total of 1196 DEG–DEG pairs were predicted in the HFs of FMD in the anagen vs. catagen group contrasts (Appendix A). In the BP category, the DEGs were enriched in the related processes of HF development and differentiation, including the regulation and processes of the hair cycle, HF development, skin epidermis development, regulation of HF development, positive regulation of HF development, skin development, the Wnt signaling pathway, epidermal growth factor receptor signaling pathway, BMP signaling pathway, epithelial cell differentiation, and epithelial cell development (Figure 3A). These DEGs involved in the key BPs were piped to STRING to generate protein–protein interaction (PPI) networks and visualized in the BPs (Figure 3B), in which hub genes from one of the MCODE model in the PPI network including *PTPRD*, ITGA2, *DAB2*, *TNC*, *SFRP1*, *CRIM1*, *DNER*, *FZD1*, *FZD9*, *KIAA1324L*, *DZIP1L*, and *TGFB2* major belonged to HF growth and development (Figure 3B). The other MCODE model containing *ITGA2*, *COL5A1*, *FBLN2*, *COL11A2*, *CRIM1*, *FGFR1*, *LAMA3*, *LGR5*, *NID1*, *TGFB2*, *PDGFA*, *PALLD*, *DST*, *SCARF2*, and *INHBA* was mainly associated with skin development and hair formation (Figure 3B). In HFs at the anagen stage and catagen stage, we further analyzed the protein–protein interaction (PPI) network of core KEGG pathways and identified 20 genes, namely *TGFB2*, *FGF13*, *TYR*, *WNT7B*, *WNT7A*, *WNT16*, *FZD1*, *FZD9*, *THBS2*, *TNC*, *ITGA2*, *ITGA5*, *SFRP1*, *NTRK2*, *INSR*, *KDR*, *CSF1*, *PDGFA*, *FGFR1*, and PDGFRA, that may participate in HF development and growth (Figure 3D).

### 3.4. Profile of sRNA Library

High-throughput sequencing of two sRNA libraries produced 36,875,130 and 82,780,595 raw reads in the HF at the anagen stage vs. catagen stage, respectively. After the QC of these raw reads, 34,128,958 (92.58%) and 74,823,520 (90.43%) clean reads were acquired (Appendix A). These clean sequences were further mapped to the FMD reference genome (Appendix A). Moreover, miRNA, tRNA, rRNA,, were identified by alignment and annotation, and the known miRNA and novel miRNA accounted for an average of 32.30% and 2.95% of non-coding RNA, and an average of 16.30% of clean sRNAs were mapped to rRNA (Appendix A). The length distributions of sRNA are shown in Appendix A. Most sRNAs ranged from 20-25 nt and accounted for at least 71.55% of the 13 samples in the HFs. A heatmap of the 13 samples of DEmiRNAs was created, which showed that DEmiRNAs and group clusters were significant (Figure 2C). Totals of 963 and 866 known miRNAs and 184 and 154 novel miRNAs were recorded at the anagen stage vs. the catagen stage, respectively (Appendix A). These novel miRNAs were determined according to their hairpin structures, precursor free energy, genome positions (Appendix A). Among them, 979 miRNAs were co-expressed in the two stages, and 168 and 41 miRNAs were peculiarly expressed only at the anagen and catagen stages, respectively (Figure 2D).

### 3.5. Functional Enrichment Analysis of DEmiRNAs and Their Target Genes

A total of 186 DEmiRNAs were identified in the HFs at the anagen stage vs. catagen stage (Appendix A), of which 33 were upregulated, and 153 were downregulated (Figure 2D, Appendix A). To recognize the possible function of the miRNAs, miRanda was used to predict the target genes of the DEmiRNAs. In total, 16,798 target genes of DEmiRNAs were predicted in the HFs at the anagen stage vs. catagen stage. We further detected the function of DEmiRNA target genes associated with HF development and growth of FMD according to GO and KEGG analysis.

In GO enrichment analysis of DEmiRNA target genes, a total of 1802 GO terms were enriched, in which the biological processes related to HF development and differentiation, including hair cell differentiation, HF development, the hair cycle process, anatomical structure morphogenesis, endothelial cell differentiation, endothelial cell development, bone trabecula morphogenesis, regulation of BMP signaling pathway, epithelial cell development, regulation of epithelial cell differentiation, epithelial cell morphogenesis, skin development, skin epidermis development, skin morphogenesis, were significantly enriched (Appendix A). KEGG analysis showed that target genes of these DEmiRNAs were enriched in 44 KEGG terms, in which the KEGG pathways related to HF development, including the Notch signaling pathway, melanogenesis, cAMP signaling pathway, cGMP-PKG signaling pathway, estrogen signaling pathway, thyroid hormone signaling pathway, phosphatidylinositol signaling system, and phospholipase D signaling pathway, were significantly enriched (Appendix A). In addition, the KEGG pathways, including the calcium signaling pathway, axon guidance, and aldosterone synthesis and secretion may participate in miRNA-mediated regulation of HF development (Appendix A).

### 3.6. Identification of mRNAs Participating in HF Development and Growth and Their Interacting miRNAs

The interaction network of DEmiRNAs and their target genes was searched using the STRING website and visualized with Cytoscape 3.8.1. Among them, 209 pairs of DEmiRNAs and their target genes were associated with HF development and growth in the Notch signaling pathway, and 269 pairs of DEmiRNAs and their target genes were correlated with HF development in the melanogenesis pathway (Figure 4 and Figure 5). In the Notch signaling pathway, miR-126, miR-196, and miR-509 were upregulated, which targeted *KAT2B*, *MAML3*, *NOTCH1*, and *PSEN2*, associated with HF growth and development, whereas miR-23, miR-24, miR-125, miR-133, miR-138, miR-149, miR-152, miR-193, miR-205, miR-210, miR-263, miR-2887, miR-29, miR-324, miR-326, miR-331, miR-34, miR-361, miR-362, miR-365, miR-370, miR-375, miR-378, miR-504, miR-542, miR-877, were downregulated, which targeted *APH1A*, *CIR1*, *CREBBP*, *CTBP1*, *CTBP2*, *DLL1*, *DLL3*, *DLL4*, *DTX1*, *DTX2*, *DVL1*, *DVL2*, *DVL3*, *EP300*, *HDAC2*, *HES1*, *HES5*, *JAG1*, *JAG2*, *KAT2A*, *KAT2B*, *MAML1*, *MAML2*, *MAML3*, *NOTCH1*, *NOTCH2*, *NOTCH3*, *NOTCH4*, associated with HF growth and development (Figure 4).

In the melanogenesis pathway, miR-9, miR-136, miR-196, and miR-509 were upregulated, which targeted *PRKCA*, *PRKCB*, *PLCB3*, associated with HF development, whereas miR-23, miR-24, miR-29, miR-34, miR-125, miR-133, miR-138, miR-149, miR-152, miR-193, miR-205, miR-210, miR-263, miR-324, miR-326, miR-331, miR-361, miR-362, miR-365, miR-370, miR-375, miR-378, miR-504, miR-542, miR-877, were downregulated, which targeted *ADCY1*, *ADCY2*, *ADCY3*, *ADCY4*, *ADCY5*, *ADCY6*, *ADCY7*, *ADCY8*, *ADCY9*, *CAMK2B*, *CAMK2D*, *CREB1*, *CREB3L1*, *CREB3L2*, *CREB3L3*, *CREB3L4*, *CREBBP*, *CTNNB1*, *FZD1*, *FZD2*, *FZD3*, *FZD4*, *FZD7*, *FZD8*, *FZD9*, *FZD10*, *WNT1*, *WNT2*, *WNT2b*, *WNT3*, *WNT3A*, *WNT5B*, *WNT6*, *WNT7A*, *WNT7B*, *WNT8A*, *WNT9A*, *WNT9B*, *WNT10a*, *WNT10b*, *WNT11*, associated with HF development (Figure 5). The above miRNAs contributed significantly to regulating target gene expression during the HF development and growth.

### 3.7. RNA-seq and miRNA-seq Data Validation

To verify the dependability of the RNA-seq, the expression profile of 11 DEGs (*WNT7A*, *WNT7B*, *WNT16*, *FZD1*, *FZD8*, *FZD9*, *FGFR1*, *NTRK2*, *SFRP1*, *ITGA5*, and *CAMK2B*) and 11 known miRNAs (let-7d, let-7e, miR-31-5p, miR-34b-5p, miR-125a-3p, miR-143-3p, miR-149-5p, miR-193a-5p, miR-324-3p, miR-362-5p, and miR-877-5p) associated with HF development was explored using RT-qPCR (Figure 6 and Figure 7). the results of RT-qPCR were fundamentally coincident with the RNA-seq data in the two stages. These results showed that our sequencing data were reliable.

## 4. Discussion

Male FMD secrete musk at the catagen and telogen stage of the HF and gradually form mature musk. During the anagen stage, the musk glands of male FMD store musk and give off a lasting aroma to attract a female for reproduction and mating. A variety of TFs, signaling pathways, and epigenetics that affect multiple cellular activities adjust and control HF development and growth. Improvements in growth and development of FMD, particularly in the male musk deer, is crucial for the breeding industry. HF development and growth are mainly regulated by DEGs, which are crucial factors affecting the development and growth of HFs. In this study, high-throughput RNA-seq was used to analyze the gene and miRNA expression profiles in the HFs during the anagen and catagen stages. The aim of this study was to identify hub genes and miRNAs involved in the growth and development of FMD HFs. This analysis helped reveal the molecular mechanism connected to HF growth and development of FMD.

It has been reported that a series of DEGs participate in the regulation of skin development, epidermis development, epidermal cell differentiation [35]. In this study, the DEGs of HF tissues of FMD at two stages were enriched by GO. It was found that HF development and growth at different stages were regulated by different genes that were involved in multiple pathways. In our study, the following pathways were enriched: melanogenesis, the Wnt signaling pathway, MAPK signaling pathway, ECM–receptor interaction, PI3K-Akt signaling pathway, Ras signaling pathway, cGMP-PKG signaling pathway, phospholipase D signaling pathway, and cytokines and growth factors. These pathways were closely correlated with HF growth and development. It has been proven that the Wnt signaling pathway is closely associated with hair growth [36]. Interestingly, the PI3K-Akt signaling pathway indirectly affects hair regeneration by promoting TGF-b2 expression [37]. In addition, the PI3k-Akt signaling pathway could increase HF stem cell differentiation by TPA treatment [38]. Recent reports indicate that the MAPK signaling pathway is activated to affect dermal papilla cell proliferation and thus regulate HF development [39,40]. Additionally, the regulatory factors in the MAPK signaling pathway influence FGF and Notch signaling pathways, which regulate HF development and differentiation [41,42]. The ECM–receptor interactions perform a dominant function in the regulation of HF size [35], whereas the Ras signaling pathway could also regulate HF morphogenesis by activating the Shh signaling pathway [43]. BMP signaling pathways participate in the normal regulation of HF development [44]. The initiation of HFs requires the involvement of a series of signaling pathways: Wnt/β-catenin, PI3K-Akt, and MAPK signaling pathways, which are associated with the epidermal cells and dermal papillae [45,46]. Our research also showed that the aforementioned signaling molecules were expressed in the HF tissues during the anagen stage of HF development.

Recent reports have revealed that miRNAs perform a regulatory function in HF development, cycling, and hair pigmentation [22,47,48]. Previous studies have indicated distinct expression patterns of certain miRNAs in different cellular compartments within the skin tissues. miR-141, miR-200, and miR-429 have been found to be highly expressed in the epidermis of skin tissues, and miR-31 is predominantly expressed in hair matrix cells [29], while the miR-199 family exhibits specific expression in the dermis of HF [49]. It is commonly known that miR-125b can inhibit stem cell proliferation via the TGF-β/BMP signaling pathway [50,51,52,53]. Our study provided the miRNA expression profile of HFs of FMD. In this study, 33 up- and 153 downregulated miRNAs were screened in the HFs at the anagen stage compared with the catagen stage. It has been reported that miR-143 [54], miR-214, miR-125b [23], miR-31 [27], and the miR-200 family [55] participate in the regulation of HF development, which is coincident with the results of our study. GO enrichment analysis of DEmiRNA target genes showed significant enrichment for biological processes involved in HF growth, development, and differentiation, including hair cell differentiation, HF development, the hair cycle, HF morphogenesis, follicle spot formation, hair cycle regulation, HF development regulation, skin development, skin epidermal development, and skin morphogenesis. KEGG analysis of DEmiRNA target genes revealed a significant enrichment of pathways involved in HF development, including the Notch signaling pathway, melanogenesis, extracellular matrix–receptor interactions, cGMP-PKG signaling pathway, phospholipase D signaling pathway, calcium signaling pathway, platelet activation, and focal adhesion. The Notch signaling pathway has been reported to regulate cell fate, proliferation, differentiation, and patterning during postnatal development of HFs [56,57]. Notch signaling is also essential for the complete maturation of HFs and postnatal hair cycle homeostasis. The ECM–receptor interaction pathway plays an important role in the regulation of HF size [58]. The ECM is widely distributed in the dermal sheath and papilla [58], the count of which determines the volume of the dermal papilla and the size of the HF [59].

A total of 169 DEmiRNAs and 236 DEGs with targeting relationships in HF tissues were discovered as a result of our combined analysis of the mRNA and miRNA transcriptome. To investigate the regulatory mechanism of miRNAs, we performed co-expression analyses of integrated DEmiRNAs and target genes to construct an interactive network. In our study, miR-125b targeted *CD28* and *ZNF541*, and miR-125a targeted *Fgf18*, *EDN1*, *DLL4*, *CD22*, *CD28*, *CD33*, *CD52*, *CD96*, *CD163*, *ZNF205*, *ZNF316*, *ZNF469*, *ZNF541*, *ZNF550*, and *ZNF784*. These members of the zinc finger family have been revealed to influence the HF cycle. During HF morphogenesis, the TF (Trps1) of ZNF was specifically expressed in the mesenchymal cell nucleus [60,61]. It has been reported that miR-29a/b1 could inhibit HF stem cell differentiation by targeting *Bmpr1a* and *Lrb6* to suppress BMP and WNT signaling [62]. In this study, miR-29b targeted *NAV1*, *MAP2K2*, *SLC7A5*, *SLC25A28*, *SLC2A4*, *SLC7A5*, *SLC25A1*, *SLC25A10*, *ZNF18*, *ZNF274*, *ZNF318*, *ZNF445*, and *ZNF142*. Several DEmiRNAs, including miR-23b-5p, miR-29b, miR-125a/b, miR-138, miR-149, miR-324-3p, miR-877, were recognized for their miRNA–mRNA interaction relationships, which may participate in HF morphogenesis (miR-23b-5p targeting *WNT6*, miR-29b targeting *PLCB1*, miR-125b targeting *CD34*, miR-138 targeting *Wnt10b*, and miR-149 targeting *Wnt7A*) and were highlighted. On the whole, these hub miRNAs may be connected to HF activity, and further study is needed to detect their functions in HF growth and development. It has been reported that miR-149-5p promotes β-catenin-induced HF stem cell differentiation in goats [63]. *CD34* is a marker of HF stem cells and HF regeneration [52]. In our research, DEG enrichment analysis revealed that *WNT16*, *WNT7A*, *WNT7B*, and *FZD1* played a key role in melanogenesis and the Wnt signaling pathway; *FZD8* and *FZD9* also played key roles in melanogenesis and the MAPK signaling pathway. *FGFR1*, *SFRP1, NTRK2*, and *ITGA5* are hub genes in cytokines and growth factors, the Wnt signaling pathway, and PI3K-Akt signaling pathway. Therefore, it is supposed that these genes perform vital functions in HF development. Subsequent validation was conducted on these genes and their targeted miRNAs. Our findings indicated that the DEGs and miRNAs identified in HFs during the anagen stage played more crucial roles in HF development compared to those identified during the catagen stage. Based on our results, it was speculated that the DEGs and DEmiRNAs of HFs at the anagen stage played important roles in HF development compared to those at the catagen stage, which was the critical period of HF initiation. Due to technical problems, the HFs of FMD were not collected at the telogen stage. Therefore, an analysis of DEGs and DEmiRNAs as well as related functions could not be performed between the anagen and telogen stages (or between the catagen and telogen stages) in the HFs of FMD.

## 5. Conclusions

In conclusion, the growth and development of HFs performs an important role in the growth cycle of HFs. In this study, RNA-seq technology was used to explore unigenes and miRNAs connected to HF development and growth in FMD. The profiles and functional analysis of DEmiRNAs and DEGs in the HFs of FMD at the anagen stage and catagen stage were contrasted, and the interaction network of DEmiRNA–mRNA was established. Unigenes and miRNAs in pathways connected to HF growth and development were differentially expressed. The DEGs and DEmiRNAs were associated with HF growth and development in the study, providing the genetic and molecular resources for the breeding of FMD. Future studies are needed to further verify the regulatory function of DEmiRNA–DEGs interactions in HF growth and development. Our findings will aid the hair control of musk deer and provide fresh thinking for musk deer farming.

## Figures and Tables

**Figure 1 animals-13-03869-f001:**
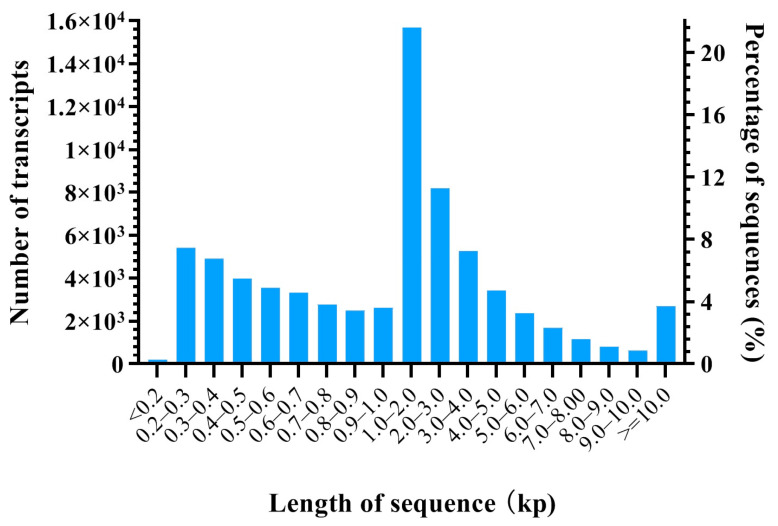
Length distribution of transcripts.

**Figure 2 animals-13-03869-f002:**
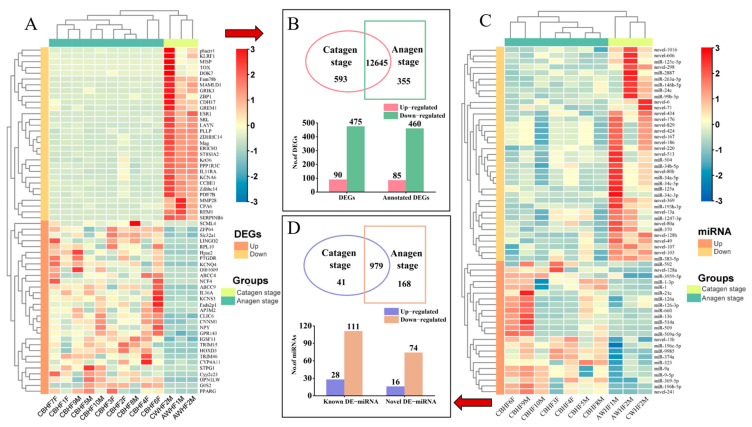
(**A**) Heatmap of DEGs. (**B**) Venn diagram of DEGs. (**C**) Heatmap of DEmiRNAs. (**D**) Venn diagram of DEmiRNAs.

**Figure 3 animals-13-03869-f003:**
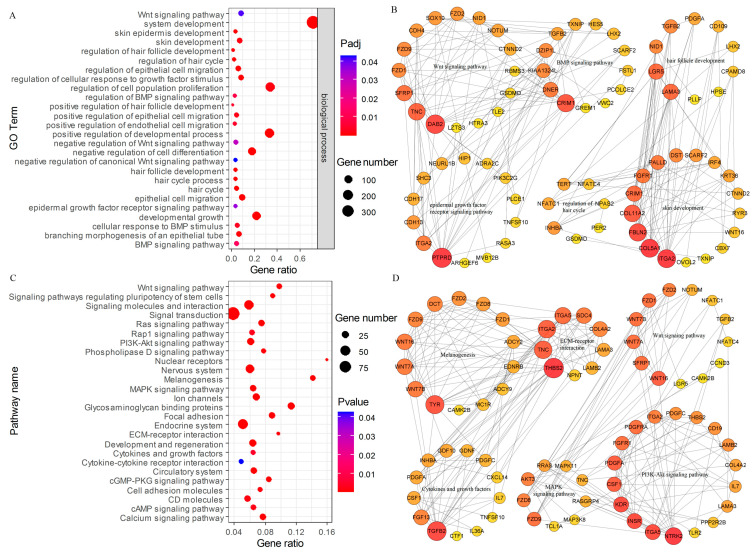
(**A**) GO enrichment associated with BP terms of HF development and differentiation. (**B**) Analysis of GO networks related to HF growth and development. (**C**) KEGG analysis of the DEGs of HFs at the anagen stage and catagen stage. (**D**) Analysis of KEGG networks related to HF growth and development.

**Figure 4 animals-13-03869-f004:**
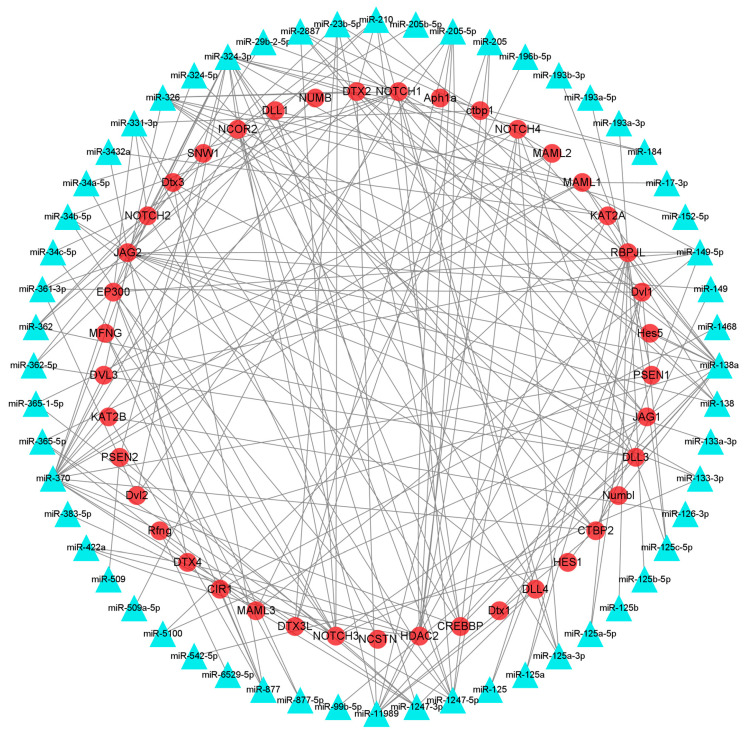
DEmiRNA and target gene interaction network related to HF growth and development of FMD in the Notch signaling pathway.

**Figure 5 animals-13-03869-f005:**
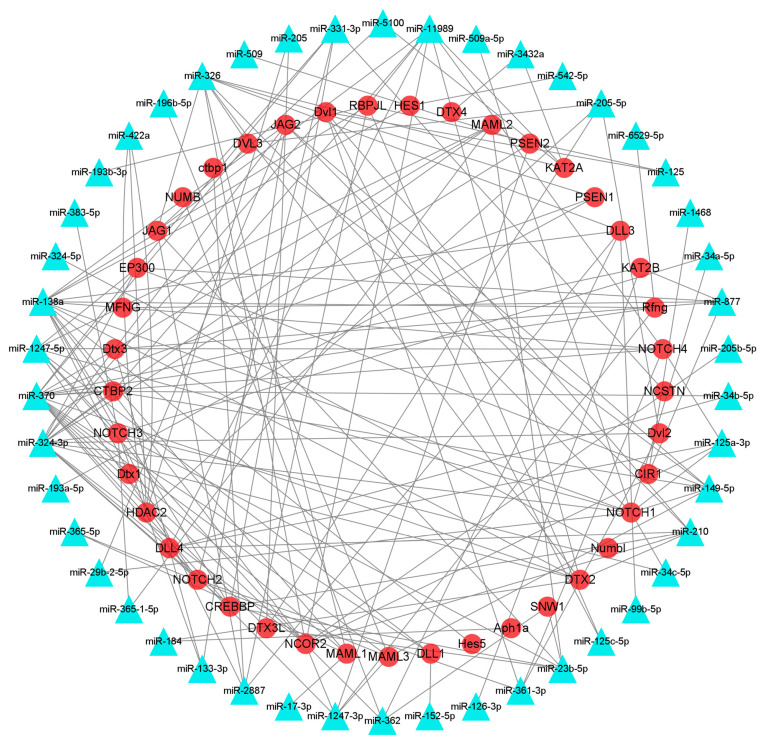
DEmiRNA and target gene interaction network related to HF growth and development of FMD in the melanogenesis pathway.

**Figure 6 animals-13-03869-f006:**
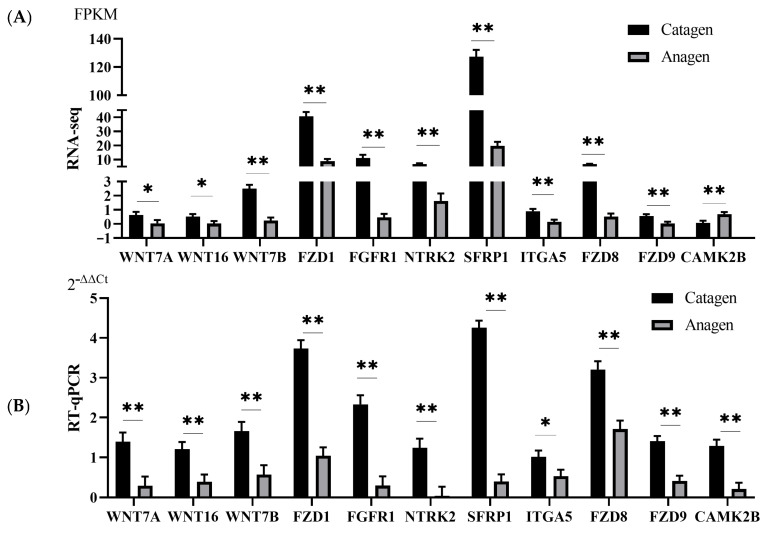
Verification of DEGs by RT−qPCR. (**A**) DEG expression in terms of FPKM as assessed by mRNA sequencing. (**B**) qRT−PCR analysis of 11 hub genes. Data represent the means ± SE. A * indicated significance; Two ** indicated extreme significance.

**Figure 7 animals-13-03869-f007:**
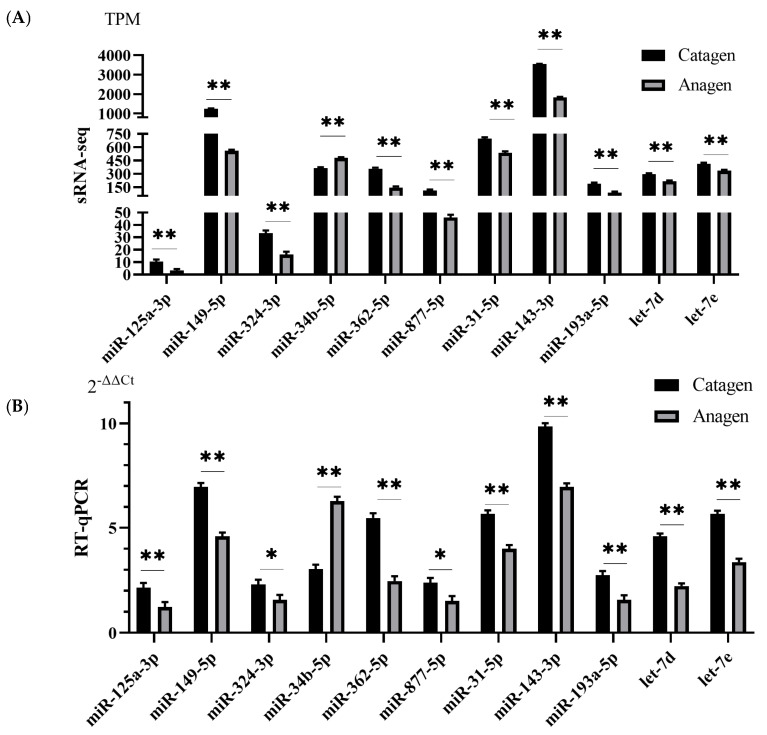
Verification of DEmiRNAs by RT−qPCR. (**A**) DEmiRNA expression in terms of TPM as assessed by RNA−seq. (**B**) qRT−PCR analysis of 11 hub miRNAs. Data represent the means ± SE. A * indicated significance; Two ** indicated extreme significance.

**Table 1 animals-13-03869-t001:** Transcripts and genes of the merged assembly.

Item	Transcripts	Genes
Number of sequences	71,167	24,352
Max length of the sequence (bp)	101,907	101,907
Min length of the sequence (bp)	132	132
Mean length (bp)	2559.9	1413.94
Total length (Mb)	182.18	34.43
Contig N50 (bp)	4942	2025
GC content (%)	49	54
≥1000 bp	41,883	11,842

## Data Availability

The data presented in this study are available in the manuscript.

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
