# Peer review of "Identification of Potential miRNA–mRNA Regulatory Network Associated with Growth and Development of Hair Follicles in Forest Musk Deer"

_animals, 2023, doi:10.3390/ani13243869_

Round 1

Reviewer 1 Report

Comments and Suggestions for Authors

The paper presents that this study systematically investigated sRNA libraries and mRNA libraries of hair follicles of forest musk deer are constructed, the libraries of which are sequenced by using Illumina HiSeq 2500, and the expression profiles of miRNA and gene in the HFs of forest musk deer are obtained at the anagen and catagen stage.  Further discoveries,the initiation of hair follicles perform an important role in the development and growth of forest musk deer, explore unigenes and miRNAs connected with HF development and growth in the forest musk deer.This has important implications for understanding the developmental mechanisms of the hair follicle in the forest musk deer, it is a topic of interest to researchers in related areas.   However, a few major issues need to be addressed.   Therefore, we believe that your paper requires revisions at this point.

Main Evaluations and Issues:

1. Please explain in detail the relationship between musk secreted by male forest musk deer and growth hair follicle cycle.

2.The collection time of skin samples was not clearly described. The anagen and the catagen are a time period, and you need to clearly specify a specific collection time point.

3. I noticed several instances where the descriptions were unclear, for example: line 239-256.This section explains which questions you want to explain and repeat the above description. Please review carefully written manuscripts and make necessary corrections to ensure that the presentation of your research is clear and accurate.

4. In the discussion section, the manuscript does not provide any direction on future research or the next steps following this study's completion. Please consider providing some guidance in this area.

5. It would be helpful if you discussed the limitations of your study more explicitly.

6. It remains unclear what the potential value of your study is. We suggest that you articulate this aspect more clearly and persuasively in the manuscript.

Author Response

For research article

Response to Reviewer #1Comments

1. Summary

Thank you very much for taking the time to review this manuscript. We have carefully evaluated the reviewers’ critical comments and thoughtful suggestions. Our manuscript was checked according to the reviewers’ comments, and the itemized response to reviewers’ comments is attached. All changes made to the manuscript are marked in red so that they may be easily identified. Thank you very much for your suggestion.

2. Questions for General Evaluation

Reviewer’s Evaluation

Response and Revisions

Does the introduction provide sufficient background and include all relevant references?

Yes

Are all the cited references relevant to the research?

Can be improved

Relevant references have been supplemented

Is the research design appropriate?

Must be improved

Modifications have been made accordingly

Are the methods adequately described?

Can be improved

Are the results clearly presented?

Yes

Are the conclusions supported by the results?

Can be improved

3. Point-by-point response to Comments and Suggestions for Authors

Comments 1: Please explain in detail the relationship between musk secreted by male forest musk deer and growth hair follicle cycle.

Response 1: Thank you for pointing this out. We agree with this comment. Therefore, we have revised and improved this section.[ line 336-338]

Comments 2: The collection time of skin samples was not clearly described. The anagen and the catagen are a time period, and you need to clearly specify a specific collection time point.

Response 2: Thank you for your suggestion, we agree with this comment. According to this comment, we have done added this part to emphasize this point. [line 121-123]

Comments 3: I noticed several instances where the descriptions were unclear, for example: line 239-256.This section explains which questions you want to explain and repeat the above description. Please review carefully written manuscripts and make necessary corrections to ensure that the presentation of your research is clear and accurate.

Response 3: Thank you for pointing this out, we agree with this comment. Therefore, we have revised and modified this section.[ line 246-253]

Comments 4: In the discussion section, the manuscript does not provide any direction on future research or the next steps following this study's completion. Please consider providing some guidance in this area.

Response 4: Thank you for your suggestions. All your suggestions are very important, which have important guiding significance for my thesis writing and scientific research. According with your advice, we have done and added this section.[ line 442-446]

Comments 5: It would be helpful if you discussed the limitations of your study more explicitly.

Response 5: Thank you for your suggestion, we agree with this comment. Therefore, we have done and added this section.[ line 430-434]

Comments 6: It remains unclear what the potential value of your study is. We suggest that you articulate this aspect more clearly and persuasively in the manuscript.

Response 6: Thank you for pointing this out, we agree with this comment. Therefore, we have revised and improved this section.[ line 110-113] 

4. Response to Comments on the Quality of English Language

Point 1: None.

Response 1: None.

5. Additional clarifications

None.

Reviewer 2 Report

Comments and Suggestions for Authors

Qui et al. provided very comprehensive analyses for exploring the regulatory network related to the growth and development of hair follicles. By combining miRNA and mRNA sequencing, the authors have identified several key genes and miRNAs related to the process. Overall, the manuscript is well-written and informative. The methods were appropriate. I have a small concern about the selection of thresholds for p values.

Minors:

Line 28: I suggest removing “obviously “ here. If the result is obvious, the authors might not need to perform this analysis.

What were the thresholds for determining the significant DEGs and DEmiRNAs? The authors should use the p values after correction for multiple testing (Line 133-139)

Line 36-38: Did the authors report the core miRNAs?

Line 136: Why did the authors use the cow reference genome, please add the reference for it as well

The reference needed for mirDeep2.

Line 299: The node names for miRNAs are not clear. The authors might increase the font size.

Why some text in the supplementary files were highlighted in different colors?

The authors should deposit the raw sequence data and might provide the raw count data for mRNAs and miRNAs in the supplementary files. 

Comments on the Quality of English Language

No comments

Author Response

For research article

Response to Reviewer#2 Comments

1. Summary

Thank you very much for taking the time to review this manuscript. We have carefully evaluated the reviewers’ critical comments and thoughtful suggestions. Our manuscript was checked according to the reviewers’ comments, and the itemized response to reviewers’ comments is attached. All changes made to the manuscript are marked in red so that they may be easily identified. Thank you very much for your suggestion.

2. Questions for General Evaluation

Reviewer’s Evaluation

Response and Revisions

Does the introduction provide sufficient background and include all relevant references?

Can be improved

Are all the cited references relevant to the research?

Can be improved

Relevant references have been supplemented

Is the research design appropriate?

Can be improved

Are the methods adequately described?

Can be improved

Specific analytical methods are presented in our manuscript

Are the results clearly presented?

Can be improved

Are the conclusions supported by the results?

Can be improved

3. Point-by-point response to Comments and Suggestions for Authors

Comments 1: Line 28: I suggest removing “obviously “ here. If the result is obvious, the authors might not need to perform this analysis.

Response 1: Thank you for pointing this out, we agree with this comment. Therefore, we have revised this section.["obviously" have been replaced by "significantly" in the revised manuscript. [line 28]

Comments 2: What were the thresholds for determining the significant DEGs and DEmiRNAs? The authors should use the p values after correction for multiple testing (Line 133-139).

Response 2: Thank you for pointing this out. Specific analytical methods are presented in our manuscript.[ line 140-142] 

Comments 3: Did the authors report the core miRNAs? [Line 36-38]

Response 3: Core miRNAs have been reported in our manuscript. [ line 324-325]

Comments 4: Why did the authors use the cow reference genome, please add the reference for it as well. [Line 136]

Response 4: We sincerely appreciate the valuable comments. We have checked the literature carefully and added references on cow genome in the revised manuscript. [Line 139]

Comments 5: The reference needed for mirDeep2. [Line 136]

Response 5: Thank you for pointing this out, we agree with this comment. Therefore, we have done and added this reference.[ line140]

Comments 6: Line 299: The node names for miRNAs are not clear. The authors might increase the font size.

Response 6: Thank you for pointing this out, we agree with this comment. Therefore, we have modified the font size of the node names for the miRNAs. [line 303,306]

Comments 7: Why some text in the supplementary files were highlighted in different colors?

Response 7: Thank you very much for careful reviews. This was due to our carelessness, and it has been corrected in the supplementary document.

Comments 8: The authors should deposit the raw sequence data and might provide the raw count data for mRNAs and miRNAs in the supplementary files.

Response 8: Thank you for your suggestions, we agree with this comment. Therefore, we have done and added this section.

4. Response to Comments on the Quality of English Language

Point 1: No comments.

Response 1: None.

5. Additional clarifications

None.
